# The Effect of pH and Sodium Caseinate on the Aqueous Solubility, Stability, and Crystallinity of Rutin towards Concentrated Colloidally Stable Particles for the Incorporation into Functional Foods

**DOI:** 10.3390/molecules27020534

**Published:** 2022-01-14

**Authors:** Ali Rashidinejad, Geoffrey B. Jameson, Harjinder Singh

**Affiliations:** 1Riddet Institute, Massey University, Palmerston North 4442, New Zealand; g.b.jameson@massey.ac.nz (G.B.J.); h.singh@massey.ac.nz (H.S.); 2School of Fundamental Sciences, Massey University, Palmerston North 4472, New Zealand

**Keywords:** flavonoid delivery systems, milk proteins, protein self-assembly, pH-driven encapsulation, ultrafiltration, functional beverages

## Abstract

Poor water solubility and low bioavailability of hydrophobic flavonoids such as rutin remain as substantial challenges to their oral delivery via functional foods. In this study, the effect of pH and the addition of a protein (sodium caseinate; NaCas) on the aqueous solubility and stability of rutin was studied, from which an efficient delivery system for the incorporation of rutin into functional food products was developed. The aqueous solubility, chemical stability, crystallinity, and morphology of rutin (0.1–5% *w/v*) under various pH (1–11) and protein concentrations (0.2–8% *w/v*) were studied. To manufacture the concentrated colloidally stable rutin–NaCas particles, rutin was dissolved and deprotonated in a NaCas solution at alkaline pH before its subsequent neutralisation at pH 7. The excess water was removed using ultrafiltration to improve the loading capacity. Rutin showed the highest solubility at pH 11, while the addition of NaCas resulted in the improvement of both solubility and chemical stability. Critically, to achieve particles with colloidal stability, the NaCas:rutin ratio (*w/w*) had to be greater than 2.5 and 40 respectively for the lowest (0.2% *w/v*) and highest (4 to 8% *w/v*) concentrations of NaCas. The rutin–NaCas particles in the concentrated formulations were physically stable, with a size in the range of 185 to 230 nm and zeta potential of −36.8 to −38.1 mV, depending on the NaCas:rutin ratio. Encapsulation efficiency and loading capacity of rutin in different systems were 76% to 83% and 2% to 22%, respectively. The concentrated formulation containing 5% *w/v* NaCas and 2% *w/v* rutin was chosen as the most efficient delivery system due to the ideal protein:flavonoid ratio (2.5:1), which resulted in the highest loading capacity (22%). Taken together, the findings show that the delivery system developed in this study can be a promising method for the incorporation of a high concentration of hydrophobic flavonoids such as rutin into functional foods.

## 1. Introduction

Rutin is a hydrophobic flavonoid derived from a wide range of natural food sources such as flowers of *Styphnolobium japonicum*, buckwheat grains, black olives, citrus fruit, asparagus, black tea, green tea, grapes, onion, plums, and elderflower tea. This flavonoid molecule, which comprises quercetin (a flavonol) and rutinose (a disaccharide), has been suggested to possess potent antioxidant properties on a molecular level. Due to its substantial radical-scavenging properties on oxidising species, this bioactive compound can show numerous therapeutic and pharmacological effects such as antidiabetic, anticancer, and anti-inflammatory [1,2,3,4]. The beneficial health effects of rutin supplementation in patients with diabetes mellitus through clinical trials have already been established [1,2,3,4]. This flavonoid has also been suggested for treating some complicated health problems such as cerebral ischemia, owing to its antioxidant properties [5,6]. Nevertheless, both pharmaceutical and nutraceutical applications of rutin are limited because of its poor aqueous solubility, which in turn, results in a low bioavailability [7,8]. Furthermore, flavonoids such as rutin (with logP of 0.15—logP is defined as the logarithm of the ratio of the concentrations of the un-ionized solute in the solvents) undergo both chemical and enzymatic degradation once exposed to either the environment (e.g., within the structure of the food or during its processing) or the gastrointestinal tract (GIT) [9,10,11].

Rutin is a glycoside consisting of the flavonol quercetin and the disaccharide rutinose (Appendix A) [12]. Only a small proportion of this flavonoid is absorbed in the intact form, because the microflora in the GIT metabolize this flavonoid to a variety of absorbable compounds such as quercetin and the aglycone rutinose [13]. Although quercetin is also a potent flavonoid with postulated health benefits similar to rutin [14,15], the level of its absorption from the GIT is also low. Accordingly, it has been suggested that increasing the solubility of rutin along with its encapsulation/protection may result in improving its bioavailability [16,17]. For this reason, there have been numerous recent attempts for delivering rutin in an encapsulated form through various delivery/encapsulation systems [18,19,20,21].

While most of these encapsulation systems decrease the degradation of rutin, they are not suitable for the delivery of this flavonoid through functional foods. This is because they often provide low encapsulation efficiency and/or loading capacity, besides using toxic solvents (e.g., chloroform) and/or complex procedures that are either expensive or difficult to scale up in the food industry [18,22,23].

Complexation of hydrophobic flavonoids (e.g., curcumin) with proteins, such as those from bovine milk (e.g., caseins), along with increasing their solubility using the pH-driven method, have already been reported [24,25]. In this regard, caseins are known to interact with hydrophobic compounds such as rutin (hydrophobic mostly due to strong self-association of phenolic moieties), owing to the presence of both hydrophobic and hydrophilic amino acids [26,27,28,29,30]. The other useful property of caseins for the delivery of hydrophobic bioactives is their dissociation (dissociation of either micellar casein or less organised casein aggregates in caseinates) under alkaline conditions and the subsequent reassociation under neutral conditions [27,31].

NaCas shares similar casein compositions with casein micelles, but it differs in amounts of calcium and phosphate [32]. Since NaCas contains both hydrophilic and hydrophobic segments, this amphiphilicity can be beneficial for the creation of self-assembled and colloidally stable particles in an aqueous medium. Furthermore, NaCas is a natural food grade biopolymer with excellent biocompatibility and digestibility, as well as its low toxicity, enabling it to be an ideal candidate for the delivery of flavonoids [33,34].

Food proteins such as caseins are nontoxic, abundant, and biodegradable. However, when used as the carrier/coating material for the delivery of hydrophobic flavonoids, often, a large ratio of protein to flavonoid is required. For example, the results of the screening experiments from our laboratory (Table 1) showed that a *w/w* ratio of 40–80:1 was required for the efficient manufacture of colloidally stable particles of rutin and NaCas. However, in the case of the stable systems containing a low concentration of both NaCas and rutin, the low concentration of solids makes the system inefficient for the delivery of rutin and its subsequent incorporation into functional food formulations. In addition, while it is known that hydrophobic flavonoids such as rutin have better solubility at alkaline pH [16,17,35], to the best of our knowledge, there is no comprehensive systematic study reporting the behaviour (chemical stability and crystallinity) of rutin under various pH conditions. Therefore, the current study aimed at studying the effect of pH (1 to 11) and NaCas addition (1.25:1 to 80:1, NaCas:rutin *w/w*) on the aqueous solubility, chemical stability, crystallinity, and morphology of rutin. This information allowed for the preparation of colloidally stable particles suitable for the incorporation of rutin into functional foods. Such a delivery system containing a high concentration of rutin, encapsulated using an organic solvent-free method, can be added to various food products such as liquids (possibly, milk-based beverages, juices, smoothies, and soft drinks) and semi-solids, in its original form, or it can be dried and added to solid functional foods.

## 2. Materials and Methods

### 2.1. Chemicals and Reagents

Rutin with a purity of >97% *w/w* (according to the manufacturer) was purchased from Sigma-Aldrich (Castle Hill, NSW, Australia). Sodium caseinate (NaCas) 180 was from Fonterra Co-operative Ltd. (Auckland, New Zealand). All other chemicals, including pepsin (porcine, 436 u/mg), and reagents used in this study were of analytical reagent grade and were obtained from either Thermo Fisher Scientific (Auckland, New Zealand) or Sigma-Aldrich (Auckland, New Zealand).

### 2.2. The Effect of pH on the Solubility, Crystallinity, and Morphology of Rutin

Rutin was mixed with Milli-Q^®^ (MilliporeSigma, Burlington, MA, USA) water at various concentrations (0.01 to 10% *w/v*) and the pH of the mixtures increased gradually (while stirring at 300 rpm) until the full dissolution was achieved. At that point, the concentration of rutin was gradually increased until it was no longer soluble. Then, the pH was increased further until the complete dissolution. This process was repeated until the maximum targeted solubility (10 *w/v*%) was achieved at pH 11. Samples were taken at each pH point (i.e., pH 1–11) and at each concentration of rutin. The solubility of rutin was calculated based on its recovery after centrifugation (3000 g, 25 °C, 10 min) of the samples and analysing the amount of soluble rutin in the supernatant using high-performance liquid chromatography (HPLC) analysis, following the method reported previously [17,35].

To study the crystallinity of rutin at each pH, samples of rutin at each pH were stored in a desiccator with silica beads (25 °C) to dry and crystallize. The dried materials were ground using a mortar and pestle and then were analysed for the degree of crystallinity at 20.0 °C using a Rigaku Rapid image-plate detector (Rigaku, The Woodlands, TX, USA) set at 127.40 mm. The Cu Kα radiation (λ = 1.540562 Å) was generated by a Rigaku MicroMax007 microfocus (rotating anode generator; Rigaku, The Woodlands, TX, USA) and an Osmic-Rigaku metal multi-layer optic device (Rigaku, USA) was used for focusing and monochromation. Samples were mounted in Hampton CryoLoops (Hampton Research, Aliso Viejo, CA, USA), and data collection was under the control of RAPID II software (Version 2.4.2, Rigaku, USA). The data were background corrected and converted to a line profile with 2DP programme (Version 1.0.3.4, Rigaku, USA), and the resulting diffractograms were compared using CrystalDiffract software (Version 6.5.5, CrystalMaker Software Ltd., Oxfordshire, UK). A 2θ angle range of 5° to 100° was used for the analysis of all the samples.

The morphology of the ground rutin crystals was studied using an environmental scanning electron microscope (SEM, FEI Quanta 200, Waltham, MA, USA) at an accelerating voltage of 20 kV. Small amounts of the samples were mounted onto aluminium stubs (using double-sided tape), sputter-coated with approximately 100 nm of gold (Baltec SCD 050 sputter coater), and then viewed under the microscope.

### 2.3. The Effect of NaCas on the Solubility of Rutin under Various pH Conditions

In order to determine the optimum amount of both rutin and NaCas to achieve the best loading, various concentrations of rutin (0.01–4% *w/v*) were dissolved in a solution of NaCas (0.2–8% *w/v*) at pH 11, which was then acidified down to pH 6, where the precipitates were formed. A duration of 15 min between each pH adjustment was allowed, and samples were taken along the process at every pH point (i.e., pH 11, 9, 8, 7, and 6; pH below 6 was not considered as the sample withdrawal was not practical due to the co-precipitation of NaCas and rutin). The withdrawn samples were left for 30 min for the complete formation of the possible precipitates. Each sample was then divided into two equal parts; the first part was centrifuged (3000× *g*, 20 °C, 10 min) in the Amicon Ultra-15 Centrifugal Filter Units containing a cellulose membrane with a molecular weight cut-off of 10 kDa (Ultracel-100K, Millipore, Burlington, MA, USA) to remove both precipitates and colloidally stable complexes, while the second part was simply filtered using a 0.45 µm membrane (Millipore) to remove the precipitates only. The concentration of rutin in the ultrafiltrates, as well as the filtrates (the second part), was determined using HPLC analysis [17], and the rutin recovery% was calculated correspondingly. As the control, the same concentration of rutin (1%) was also treated and analysed the same way as the rutin–NaCas mixture. In this study, for simplicity, the recovery % of rutin is referred to as its solubility.

### 2.4. Formation of the Colloidally Stable Systems (Rutin–NaCas Particles)

The ratio of 2.5:1 (NaCas:rutin *w/w*) was selected due to the best loading capacity achieved (Table 2; Formulation UF4). This was decided based on the results of the systematic screening of various formulations of NaCas and rutin and on the understanding of the solubility and stability of rutin at different pH values. The effect of NaCas (at various concentrations) on the solubility of rutin and the physical stability of rutin–NaCas complexes at different ratios and conditions (explained in the previous sections) were measured. Therefore, rutin was dissolved in a NaCas solution at pH 11, and the solution was mixed (300 rpm) for 30 min and then heated to 80 °C. The neutralization of the rutin–NaCas solution was carried out using HCl (0.1–1 M), and the mixture was high-shear mixed (33,000 rpm, 3 min, 20 °C). The excess water was then removed using ultrafiltration (Amicon^®^ Ultra-15 Centrifugal Filter Units; Millipore, USA), and the dispersions were stored at 4 °C until further analysis. The colloidal stability was assessed based on the data obtained from particle size and zeta potential analyses, as explained in the next section.

### 2.5. Particle Size and Zeta Potential Analyses

A Malvern Zetasizer Nano (Malvern Instruments Ltd., Worcestershire, UK) was used for measuring the particle size of particles smaller than 600 nm and the surface charge of all particles (Malvern Instruments Ltd., Malvern, UK). Size of the particles bigger than 600 nm (identified during the screening experiments) was measured by a Mastersizer (Malvern Instruments Ltd., Malvern, UK). The samples (pH 7) that were required to be analysed by the Zetasizer were diluted (1:16) in Milli-Q water.

### 2.6. Encapsulation Efficiency (EE) and Loading Capacity (LC)

To measure the amount of rutin encapsulated inside NaCas particles (EE), the samples were centrifuged (3000× *g*, 20 min, 22 °C). First, to release the total fraction of rutin, the supernatants were disrupted in heated ethanol (70 °C; 1:1 *v*/*v*) [36] prior to their filtration using a 0.45 µm membrane filter (Thermo Scientific, Waltham, MA, USA). Based on our previous study [17], rutin is soluble in ethanol at a concentration of about 4% *w/v*. The concentration of rutin in the supernatant was then determined by HPLC analysis using a developed isocratic method, based on a previously published method [35] with slight modification. The HPLC machine was equipped with UV–Visible and diode-array detectors (Agilent Technologies, 1200 Series, Santa Clara, CA, USA) and a reverse-phase Prevail™ C18 column (4.6 cm × 150 mm and 5 μm particle size; Grace Alltech, Columbia, MD, USA). Acidic Milli-Q water (pH 3.50, 1% acetic acid) and methanol at a volume ratio of 50:50 were used for the mobile phase that was pumped at a flow rate of 1 mL/min (sample injection volume of 5 µL). Rutin was detected at the wavelength of 356 nm at a retention time of about 4.8 min. Standard solutions (0.01–1 mg/mL) of pure rutin (>97%) in the mobile phase were used for the calibration of the HPLC column and plotting the standard curve, before the quantification of rutin in the samples. The chromatographic peaks of analytes were identified and quantified by comparison of retention times with the rutin standard and by peak integration using the external standard method.

Finally, the encapsulation efficiency (EE) of rutin was calculated using the following formula:EE (%) = (C_sup_/C_total)_ × 100(1)
where C_total_ is the total (initial) concentration of rutin (*w/v*) in the system, and C_sup_ is the rutin concentration (*w/v*) in the supernatant phase of the system. Loading capacity (LC) was calculated using the following equation:LC (%) = (C_sup_/weight of the particles (after drying)) × 100(2)

### 2.7. Release Kinetics of Rutin during the Simulated Gastric Digestion

Control NaCas or rutin–NaCas mixtures were mixed with a fasting solution (containing 3.2 mg/mL pepsin) and fed into the human gastric simulator (HGS) at 37 °C. Notably, control rutin could not be used for this part of the study due to its insolubility in water (under digestion conditions), leading to its precipitation at the bottom of the digestion unit.

100 g of the sample was mixed with 12 mL fasting solution, and this was fed into the HGS. The conditions were set as explained in a previous publication [37]. Then, the fasting solution was pumped into the HGS at a rate of 0.6 mL/min, while the pepsin (16 mg/mL) was pumped into the HGS at a rate of 0.15 mL/min. The experiment was run for 180 min.

For the accurate control of the gastric emptying, 15 mL of digesta was removed out from the stomach every 20 min, equalling the gastric emptying rate of 0.75 mL/min. The contraction frequency was 3 times/min, simulating the actual contraction of the human stomach. A heater equipped with a thermostat was used for maintaining the temperature of the HGS at 37 °C throughout the experiment. The maximum digestion time was 180 min, with the samples taken at intervals of 20, 40, 80, 120, and 180 min, with 0 min sample being the control. The initial pH in the HGS was defined as the pH of the dispersion. With the ingestion of the fasting solution (0.6 mL/min pH 1.5) and gastric emptying of 15 mL/20 min, the pH in the HGS at different times was assumed to be that of the emptied digesta because the set-up (roller contraction) prevented easy access into the HGS [37]. The digesta taken at different time points were sieved with a 1 mm pore-sized mesh and dried at 105 °C overnight in a vacuum oven to obtain the dry weight for each sample from different intervals. The protein content in the aqueous phase during time-dependent hydrolysis by pepsin was determined using the Biuret assay [38].

### 2.8. Statistical Analysis

All samples were prepared in triplicate, and all measurements were repeated three times. Mean values of data and standard deviations were calculated using Excel 2016 (Microsoft Redmond, VA, USA), and the significant differences between treatments were evaluated using SPSS 20 Advanced Statistics (IBM, Armonk, NY, USA). One-way analysis of variance (ANOVA) was performed with the Tukey’s multiple-comparison test at *p* < 0.05 for the mean comparison.

## 3. Results and Discussion

### 3.1. The Effect of pH on Aqueous Solubility, Stability, and Crystallinity of Rutin

Different concentrations of rutin (0.01–5% *w/v*) were dispersed in water, and after 30 min, the dispersions were centrifuged, and the concentration of rutin in the supernatant was determined (Figure 1). At a low concentration (0.01% *w/v*), most of the rutin was soluble in water regardless of the pH, which corresponds with the results of our previous study where the aqueous solubility of the same rutin product was evaluated [17], as well as the results of other studies [1,39].

In the case of the dispersion containing 5% rutin, most of the rutin (about 90%) was soluble at pH 9, but the solubility started to decrease below this pH (Figure 1). This was expected, as a pH below 9.0 is outside the major p*K*_a_s of rutin (ranging from 7.1 to 11.65) [40,41]. This is in agreement with the results of another published report [42], where a slight degradation of rutin was found at pH 11.0 at 21 °C, over a 30 min time period, which could be attributed to the decomposition of this flavonoid into phenolic acids under an alkaline environment (i.e., pH 11). Buchner [43] reported that although rutin concentration remained almost unchanged when kept under weak acidic conditions (pH 5), it depleted to approximately 20% after 5 h when the pH was constantly adjusted to 8. Additionally, similar to many other flavonoids, oxidative degradation of rutin is also possible [43,44]. Not only do flavonoids such as rutin degrade if kept at alkaline pH for a prolonged period, but the products of alkaline nature are not desirable for the incorporation into most food products since they can result in undesirable changes in sensorial and organoleptic properties of such products.

We studied the crystallinity and morphology of the fabricated crystals by growing rutin crystals under controlled humidity (desiccator; see Section 2.2). The XRD data presented in Figure 2 revealed that the dried rutin particles at pH 11 and 10 were amorphous, but small crystals started to grow at pH 9. However, at pH < 8, all particles were in crystalline form, while at pH close to 1.0, crystalline rutin was not present. The change in rutin crystallinity over this pH range may be associated with the protonation state of the various phenolic OH groups in the rutin molecule, although for strongly acidic conditions, loss of rutin crystallinity could possibly be attributed to oxidative damage [45]. Based on the ^13^C cross-polarization magic-angle spinning (CP-MAS) nuclear magnetic resonance (NMR) spectroscopy of quercetin and its monosodium salt [45], the signals of carbon atoms C1′−C6′ in ring B of neutral quercetin were reported to overlap with the C1′−C6′ signals for its monosodium salt. For the monosodium salt, the chemical shift for C-3 in ring A was unchanged compared to neutral quercetin, whereas the signal for C-7 was shifted upfield, indicating the site of phenol deprotonation [45]. The rutin samples that showed crystalline diffraction all shared the same diffraction pattern, indicating a common chemical composition that was attributed to neutral rutin.

The crystallinity results obtained for rutin at various pH conditions in the current study were confirmed using SEM (Figure 3). At pH 11, 10, and 9, rutin particles shared a common arrowhead-like morphology that was distributed throughout the specific micrographs. The crystals that formed at the pH close to the neutral pH were more granular and similar to those previously reported [17,39,46]. At pH 9, both morphologies appeared to be present. The particles formed at pH 1 were less crystalline than those formed at 2 ≤ pH ≤ 8, which is in close agreement with crystallinity data shown in Figure 2. The surprisingly low intensity for crystalline rutin at pH 5 and 6, compared to pH 4 and 7, is attributed to imprecision in placing identical masses of sample in the nylon loop. Possibly, at pH 1, 10, and 11, rutin is not in the crystalline form. Thus, based on these observations, it is confirmed that rutin is highly soluble at alkaline pH, but its solubility decreases dramatically at pH < 9; nonetheless, its behaviour under such conditions in terms of chemical stability is yet to be fully understood.

Paczkowska [47], in a study that aimed to modify the properties of rutin (e.g., chemical stability, solubility, antibacterial activity, dissolvability, and permeability), reported slow decomposition of rutin in 0.2 M NaOH with or without beta-cyclodextrin (even slower in 0.5 M HCl), and a small increase in solubility on binding to beta-cyclodextrin.

### 3.2. The Effect of NaCas on Solubility and Stability of Rutin

In order to understand the effect of NaCas on the solubility of rutin at different pH values, we carried out a separate experiment at a constant ratio of NaCas:rutin (1:1 *w/w*). A solution of rutin of the same concentration (i.e., 1% *w/v*) was considered as the control. As shown in Figure 4, there was substantially less rutin recovered from the rutin–NaCas mixture than from the rutin solution by itself in the pH range from 11 to 7, which might indicate a degree of association between the two compounds (i.e., NaCas and rutin). However, there was a dramatic decrease in rutin solubility at pH < 8.0 for both the control rutin and the rutin–NaCas mixture. This was due to the precipitation of rutin at pH < 8.0, as explained earlier (Section 3.1, Figure 1), as well as in our previous publications [16,17].

There were low recoveries observed for rutin in both samples (i.e., control free rutin and rutin–NaCas) at pH 6.0 and below. As mentioned earlier (Section 3.1), although it was possible to dissolve 10% (*w/v*) rutin at pH 11.0, dramatic precipitation of 5% *w/v* rutin at pH < 8.0 was apparent (Figure 1) due to the substantial decrease in the solubility of rutin, due to protonation of phenolate groups (Figure 2; Figure 3). This was also observed visually, where the rutin precipitates and/or rutin–NaCas co-precipitates could be seen at pH < 8.

We also determined the chemical stability of rutin (1% *w/v*) using HPLC analysis: when rutin was dissolved at pH 11.0 and kept at this pH for 30 min, there was about a 10% decrease in its initial concentration. The degradation of rutin under alkaline conditions may be associated with the decomposition of rutin into small phenolic acids and its derivate quercetin [48]. As mentioned in Section 1, quercetin is also a potent flavonoid with suggested health benefits similar to rutin [14,15]; although similar to rutin, the level of its absorption from the GIT is also low. At pH < 6, the solubility curves for both control rutin and rutin–NaCas solutions were essentially baselined, confirming that almost all rutin was precipitated at this pH regardless of its association with NaCas (Figure 4).

Although the interactions between NaCas and rutin have not received much attention, the binding behaviour of quercetin (the aglycone rutin derivative) with β-casein has been studied before [24], and it has been reported that quercetin could form a 1:1 complex with β-casein, possibly due to both hydrogen bonding and van der Waals interactions [24]. The molecular docking studies [24] also suggested the hydrophobic core of β-casein as the binding site for quercetin, where the quercetin molecule made five hydrogen bonds and a host of van der Waals contacts with hydrophobic side chains and charged amino-acid side chains. Another important finding [24] was that the molecular dynamics simulation suggested that quercetin could interact with β-casein, with little effect on β-casein’s rather minimal secondary structure elements but with significant compaction of the protein molecule on binding quercetin. Altogether, the results presented in this section (Figure 4) confirm that such interactions may exist and that such an approach can be used for the delivery of rutin; i.e., the association of rutin and NaCas in a controlled manner can lead to better solubility and stability of rutin at pH 7–8.

### 3.3. Characteristics of the Concentrated Rutin–NaCas Particles

Table 1 shows the formulations that were screened to determine the best NaCas:rutin ratio in terms of size, surface charge, and physical stability of the particles at neutral pH. The visual appearance of some of these formulations can also be seen in Appendix A. While some of the formulations presented in Table 1 produced stable dispersions (where no phase separation or precipitation was observed) with a highly negative surface charge (i.e., colloidally stable formulations), the NaCas:rutin ratio in such formulations was comparatively high. Some of the stable formulations (including Q, T, W, and Z) were selected for further processing (i.e., concentration by ultrafiltration) in order to improve the loading of rutin into NaCas.

The particle size of the concentrated formulations (Table 2) varied from 157 to 230 nm and the zeta potential from −13.5 to −38.7 mV, depending on protein:flavonoid ratio, which is in agreement with the results previously reported [25,43]. In general, when the control sample (i.e., containing only NaCas) was compared with the sample containing rutin–NaCas particles, the presence of rutin increased the particle size, leading to increased particle surface area by a factor somewhat less than 2, while also substantially shifting the surface charge of the particles to more negative values by a factor of >2 (except for UF1C and UF1). This would indicate that rutin not only influences particle size but also influences surface charge on the particles. Other researchers [43] have reported that the interactions between the cationic amino acid residues of casein and carboxylate groups of pectin resulted in the formation of larger particles when compared with the control NaCas.

The encapsulation efficiency and loading capacity of rutin in the concentrated systems were about 76–83% and about 2–22%, respectively. The visual appearance of the final systems containing high concentrations of rutin can also be seen in Appendix A. Thus, the formulation containing 5% NaCas and 2% rutin was chosen due to the ideal protein:flavonoid ratio (2.5:1), which resulted in the highest loading capacity (about 22%). The particles manufactured in the present study exhibited greater values for both EE and LC when compared with those reported in the previous studies for a different matrix for rutin [49,50,51].

Caseins have already been suggested as delivery vehicles for hydrophobic bioactive molecules due to their ability to self-assemble or co-assemble to form some supra-structures [43,52,53]. As caseins contain a high content of proline, they therefore exist in the form of an open structure, and they are highly accessible for proteolytic cleavage in the digestion tract. These special properties of caseins can be used as a mechanism for the target release/delivery of various hydrophobic compounds such as hydrophobic flavonoids [52]. The particles in selected mixtures of rutin and NaCas are colloidally stable, which indicates some degree of the complexation between rutin and NaCas. Using dynamic light scattering and ultracentrifugation, the dissociation of NaCas at high alkaline pH and its subsequent reassociation after decreasing the pH to close to neutral has been previously confirmed [35]. Such consequent neutralization of NaCas was used for the encapsulation of curcumin in colloidally stable self-assembled particles of casein [35]. Although the system developed in the current study is based on the self-assembly of the NaCas, which was previously used for the delivery of curcumin [35], we have overcome the problem of poor loading of rutin through the systematic investigation of its aqueous solubility under various pH conditions and its interaction with NaCas, possibly through the molecular interactions between rutin and NaCas [24,54], besides the effect of rutin on the potential cross-linking of casein molecules. Therefore, this system, with a high concentration (mass ratio of 2.5:1, NaCas:rutin *w/w*) of rutin, may be able to release rutin efficiently in the digestive tract.

### 3.4. In Vitro Digestion of the Rutin–NaCas Complexes and Rutin Release Behaviour

The pH profile, particle size, and the appearance of rutin–NaCas particles (NaCas:rutin ratio of 2.5 *w/w*) vs. control NaCas throughout the simulated gastric digestion are shown in Figure 5. Both control NaCas and rutin–NaCas complexes exhibited the same pH profile at the early stage of gastric digestion, but this behaviour changed after around 80–120 min of digestion such that the sample containing rutin resulted in a lower pH. Hydrolysis of caseins by pepsin at acidic conditions increases pH due to the formation of a basic carboxylate moiety that picks up protons upon hydrolysis of peptide bonds. If this process is retarded, as in the rutin–casein complex, then the pH decreases more rapidly as more of the highly acidic SGF is added in the simulated digestion. Figure 5A shows this retarded hydrolysis for the rutin–casein complex. This is confirmed in Figure 6A, which shows that protein concentrations at times longer than around 40 min remained higher in the case of the rutin–casein complex than for sodium caseinate alone. Effective protection of casein by rutin against hydrolysis was provided at the mid-to-late stages of digestion, as is apparent in Figure 6A. Rutin is subsequently released during the second phase of digestion (where the complexes reach the small intestine). This then makes the NaCas–rutin species, compared to rutin alone, more favourable in terms of the oral delivery of this flavonoid.

The particle size analysis of the samples (including both phases and immediately after the withdrawal) during the gastric digestion (Figure 5B) showed that both control NaCas and rutin–NaCas complexes followed a similar pattern in terms of the increase in particle size, with particle size increasing initially before decreasing at longer times beyond 100 min. This may also be another indication that the rutin–casein complexation can result in delaying the release of rutin, as well as the aggregation of the protein (i.e., NaCas). This can also be seen from the pictures presented in Figure 5C, where the aggregation of the protein particles in the absence of rutin (Figure 5C1) can be observed directly after 20 min, while such aggregation could not be seen in the case of rutin–NaCas complexes until the sample was digested for at least 40 min. Therefore, in the case of rutin–NaCas complexes, the precipitation of protein was less than the control NaCas, which may be attributed to the possible interactions between rutin and caseins [43,55], as explained in the previous sections. Nevertheless, as seen in Figure 5C, after 120 min and in the case of the rutin–NaCas sample, such aggregations start to disappear, which agrees with the decrease in the particle size data reported in Figure 5B and the increasing extent of hydrolysis apparent in Figure 6A.

Although there are no systematic data available for the rutin–casein complexations thus far, such complexations between other polyphenols and milk proteins are known to occur [56,57,58,59]. Zhao [58] studied the interactions between proteins such as casein and phenolic compounds such as tannic acid and gallic acid and the effect of such interactions on the properties of proteins. Phenolic compounds changed the structure of the proteins that lead to a significant decrease in their digestibility and reduced the degree of protein hydrolysis during gastrointestinal digestion [58]. Additionally, it is also known that some polyphenols may not only bind onto different sites of proteins at their low concentrations, but can also covalently cross-link protein molecules when present at higher concentrations [59,60,61]. Rawel [59] confirmed the interactions between whey proteins and both quercetin and rutin that resulted in blocking the lysine, tryptophan, and cysteine residues. These researchers [59] reported that the phenolic reactant was covalently bound to a β-lactoglobulin (β-Lg) molecule, while the fractions of high molecular protein were also detected by polyacrylamide gel electrophoresis (SDS-PAGE), possibly due to the cross-linking of β-Lg with quercetin. It has also been suggested that compared to smaller phenolic compounds (e.g., phenolic acids), the polyphenolic compounds with higher molecular weight can make stronger cross-links with proteins. This is due to the presence of several aromatic rings on these molecules, meaning that there are more sites for the possible reactions to take place [60]. In a previous study [61], polyphenols have also been suggested as cross-linkers of protein-based products such as gelatin gels and gelatin-based coacervates for use as novel ingredients in the food industry. These researchers [61] observed that such a cross-linking led to denser polymeric networks that prevented the possible extension of the peptide chains at the pH away from the isoelectric point.

Furthermore, we studied the protein hydrolysis behaviour of both control NaCas and rutin–NaCas complexes (Figure 6A) and found that the slope of protein hydrolysis in the case of control NaCas was steeper than the rutin–NaCas complexes. After 40 min of digestion, the protein concentration curve of the control rutin started to show a steady linear decrease to about 10 mg/mL, whereas in the case of the rutin–NaCas particles, such a decrease was not seen until the end of the digestion period.

Such a hydrolysis/release profile may protect rutin from early release due to the lower rate of casein hydrolysis in the case of rutin–NaCas particles when compared with the hydrolysis rate of caseins in NaCas control. This slower hydrolysis may partially be owed to the hydrophobic interactions between NaCas and rutin, indicating that the self-assembled NaCas particles can be appropriate for the delivery of hydrophobic flavonoids such as rutin [25,31]. Most protein-based particles developed for the oral delivery of flavonoids such as rutin are easily degraded in the stomach by proteases, which in turn leads to the rapid release of the encapsulated flavonoid before it arrives in the small intestine [62,63]. This makes the applications of such delivery systems limited, a challenge that we have addressed in this current research, where the encapsulation of rutin by NaCas retarded the hydrolysis of casein and consequent release of rutin (to then precipitate out under acidic conditions). In other words, this delivery system can protect rutin under the acidic condition of the upper part of the digestion system (i.e., stomach) and release it under the neutral condition of the lower part. Moreover, the digestion of rutin–NaCas species could be achieved with a higher total concentration of rutin than in the rutin alone (experiment control).

## 4. Conclusions

Based on the systematic assessment of the aqueous solubility of rutin and screening various formulations of NaCas and rutin (i.e., controlling the mass ratio of protein and flavonoid) for the development of an efficient oral delivery towards rutin incorporation into functional foods, the delivery system containing 5% NaCas and 2% rutin (the NaCas:rutin ratio of 2.5:1), prepared at alkaline pH and brought to neutral pH, is considered as the most efficient system. This system also protected rutin under the simulated gastric digestion, with a limited release of this flavonoid under these conditions, owing to the successful complexation of rutin and NaCas, suggesting that it is a promising method for the incorporation of high concentrations of rutin into functional foods (in particular, liquid and semi-liquid products). Although there appear to be numerous encapsulation systems suggested for the oral delivery of rutin, most of these systems fail at addressing the required amount of rutin (500 mg/d) to be delivered in a single dose of the final functional food product, due to the poor aqueous solubility of this bioactive flavonoid that results in the poor loading capacity of the corresponding delivery systems. Thus, the development of the rutin delivery system reported in this study, using NaCas as a conventional and inexpensive food grade protein, can pave the way for a more feasible delivery method for the incorporation of rutin into functional food products. Yet, the improved stability of such particles and higher protection of rutin may also be further achieved if biopolymers such as gum arabic, maltodextrin, and pectin are used as a component of this system [43,64,65]. We are currently investigating the potential changes in the protein composition of samples and the structure of the protein as a function of digestion time. Further research is also currently being carried out in our laboratory to explore the release behaviour of rutin from the rutin–NaCas particles beyond the gastric phase, as well as the possibility of using proteins (from both animal and plant sources) for the delivery of various hydrophobic flavonoids for their incorporation into various functional food products.

## Figures and Tables

**Figure 1 molecules-27-00534-f001:**
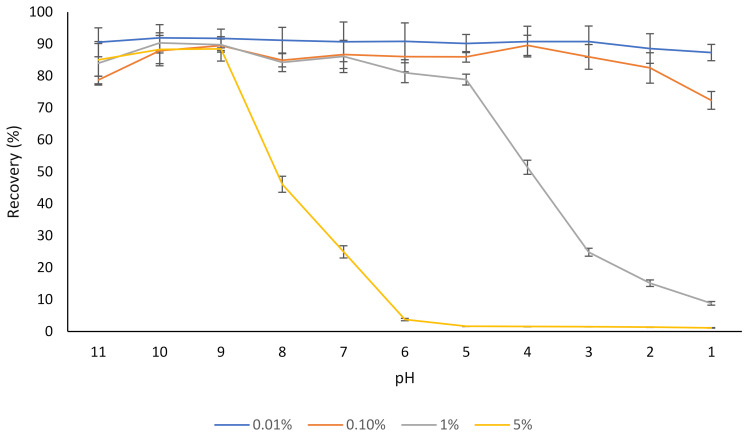
The effect of pH on the aqueous solubility (recovery) of different concentrations of rutin (0.01–5%). The results are means of three replicates of measurements.

**Figure 2 molecules-27-00534-f002:**
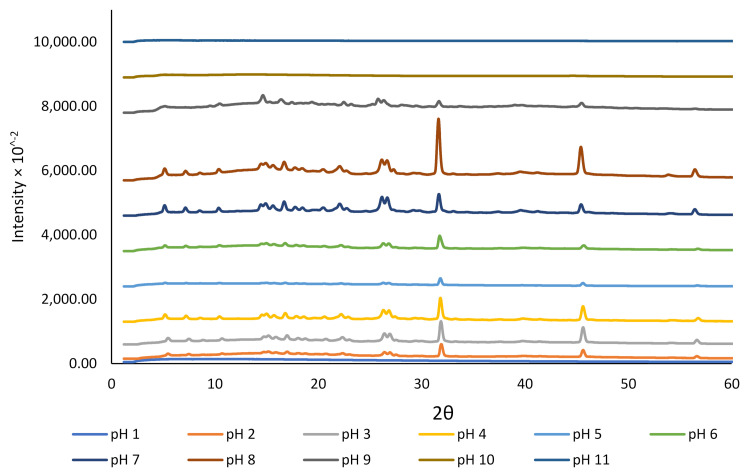
The effect of pH on the crystallinity of rutin encapsulated in NaCas (1%).

**Figure 3 molecules-27-00534-f003:**
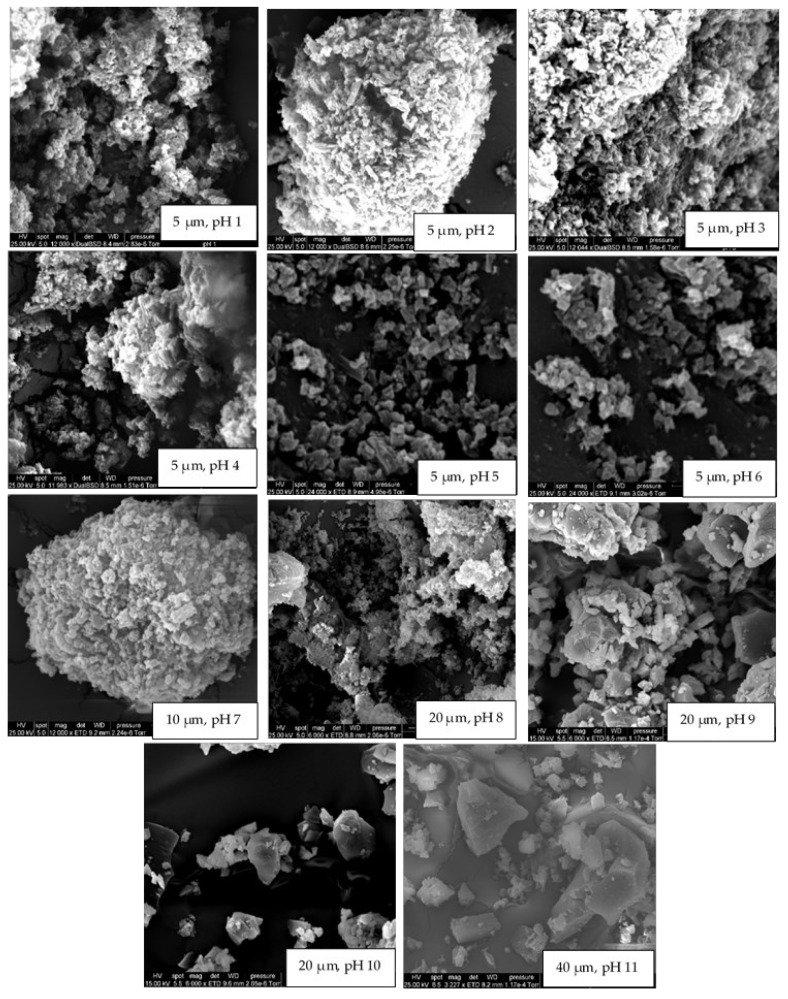
The effect of pH on the morphology (scanning electron micrographs) of rutin crystals grown under controlled humidity. pH values and scale bars can be found at the bottom of each micrograph.

**Figure 4 molecules-27-00534-f004:**
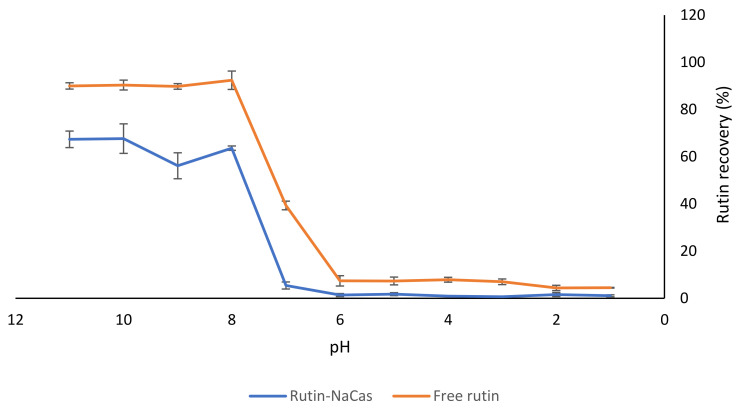
The effect of NaCas on the recovery of rutin from the NaCas–rutin mixture (1%). The results are means of three replicates of measurements.

**Figure 5 molecules-27-00534-f005:**
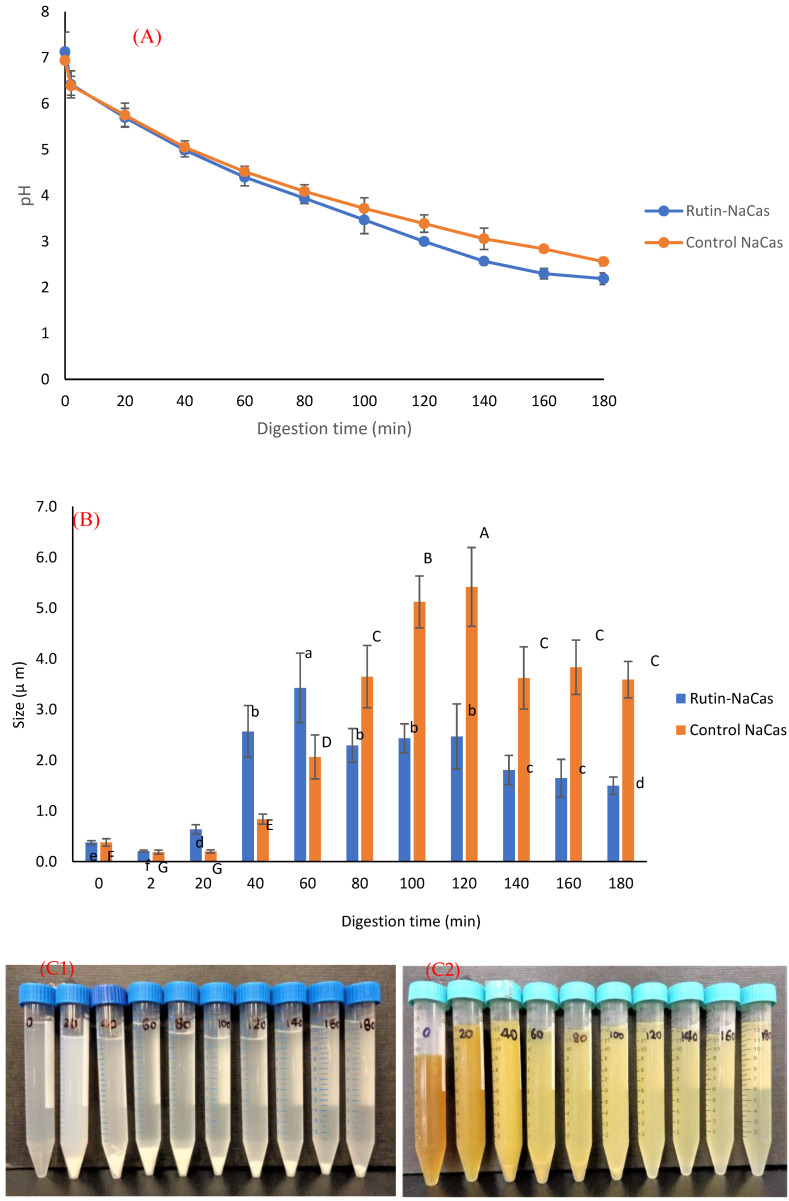
The behaviour of control NaCas and rutin–NaCas nanocomplexes (NaCas:rutin ratio of 2.5 *w/w*) throughout the simulated gastric digestion. (**A**) pH profile; (**B**) particle size; (**C**) appearance (C1: control NaCas, C2: rutin–NaCas). Columns of the same colour containing different letters are significantly different at *p* < 0.05.

**Figure 6 molecules-27-00534-f006:**
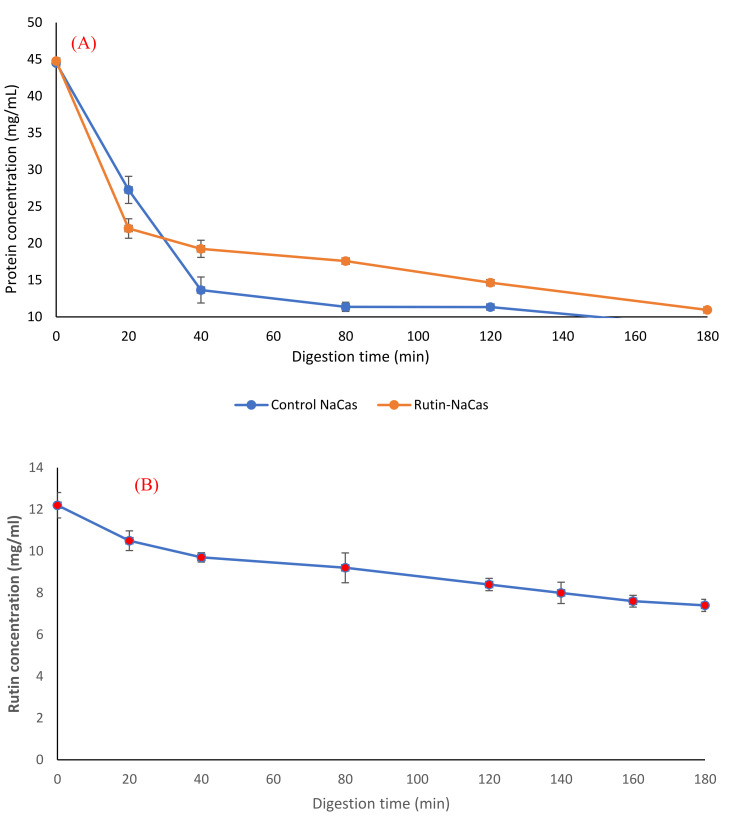
Protein release behaviour (**A**) and rutin release behaviour (**B**) of the rutin–NaCas nanocomplexes during the simulated gastric digestion.

**Table 1 molecules-27-00534-t001:** The effect of sodium caseinate (NaCas):rutin ratio on the size, surface charge, and stability of the particles in the encapsulation system (pH 7).

Formulation	NaCas Concentration (% *w/v*)	Rutin Concentration (% *w/v*)	NaCas: Rutin Ratio (*w/w*)	Size (nm)	Zeta Potential (mV)	Colloidally Stable?
A	5	1	5	1880 ± 50 ^b^	−21.3 ± 2.2 ^a^	No
B	5	2	2.5	2120 ± 40 ^a^	−22.8 ± 1.7 ^a^	No
C	5	4	1.25	1850 ± 30 ^b^	−23.0 ± 1.5 ^a^	No
D	1	0.2	5	1380 ± 40 ^f^	−22.8 ± 1.4 ^a^	No
E	1	0.4	2.5	145 ± 70 ^e^	−23.6 ± 2.6 ^ab^	No
F	1	0.8	1.25	1490 ± 40 ^d^	−24.6 ± 1.8 ^c^	No
G	1	0.1	10	832 ± 22 ^h^	−22.6 ± 1.2 ^a^	No
H	5	0	NA	214 ± 5 ^qp^	−30.2 ± 1.5^d^	Yes
I	1	0	NA	203 ± 8 ^r^	−31.2 ± 2.3 ^de^	Yes
J	2	0	NA	259 ± 14 ^o^	−29.7 ± 1.0 ^d^	Yes
K	2	0.1	20	1520 ± 60 ^d^	−24.2 ± 1.03 ^c^	No
L	2	0.2	10	1670 ± 60 ^c^	−25.31 ± 1.1 ^c^	No
M	2	0.4	5	1400 ± 80 ^f^	−22.4 ± 1.4 ^a^	No
N	2	0.8	2.5	1320 ± 70 ^g^	−21.1 ± 1.0 ^a^	No
O	2	1.6	1.25	1740 ± 40 ^i^	−23.5 ± 1.3 ^ab^	No
P	0.2	0	NA	172 ± 17 ^s^	−24.5 ± 0.9 ^c^	Yes
Q	0.2	0.01	20	236 ± 21 ^p^	−23.1 ± 0.9 ^b^	Yes
R	0.2	0.02	10	321 ± 27 ^n^	−22.7 ± 1.2 ^a^	Yes
S	0.2	0.04	5	450 ± 30 ^k^	−21.3 ± 1.6^a^	Yes
T	0.2	0.08	2.5	460 ± 40 ^k^	−23.5 ± 1.4 ^ab^	Yes
U	0.2	0.16	1.25	620 ± 40 ^j^	−22.7 ± 1.4 ^a^	No
W	8	0.2	40	362 ± 17 ^m^	−25.0 ± 1.0 ^c^	Yes
X	8	0.1	80	422 ± 18 ^l^	−24.8 ± 0.8 ^c^	Yes
Y	4	0.1	40	445 ± 29 ^kl^	−30.4 ± 2.0 ^d^	Yes
Z	4	0.05	80	271 ± 22 ^o^	−31.4 ± 2.2 ^de^	Yes
WC	8	0	NA	226 ± 10 ^p^	−32.7 ± 1.2 ^e^	Yes
YC	4	0	NA	220 ± 6 ^p^	−30.3 ± 0.9 ^d^	Yes

Note: the results are means of three replicates of measurements. NA: the measurement was not applicable. The means within the same column containing different superscripts are significantly different (*p* < 0.05).

**Table 2 molecules-27-00534-t002:** The properties of the concentrated rutin-sodium caseinate (NaCas) nanoparticles (pH 7).

Formulation	NaCas Concentration (%)	Rutin Concentration (%)	NaCas: Rutin Ratio (*w/w*)	Size (nm)	Zeta Potential (mV)	Encapsulation Efficiency (EE, %)	Loading Capacity (LC, %)
UF1C	66.81	0	NA	218 ± 7 ^b^	−26 ± 3 ^b^	NA	NA
UF1	66.81	1.67	40	208 ± 5 ^c^	−38.7 ± 1.5 ^d^	83 ± 4 ^a^	2.03 ± 0.06 ^d^
UF2C	39.64	0	NA	167 ± 13 ^e^	−13.5 ± 0.9 ^a^	NA	NA
UF2	39.64	1.98	20	230 ± 4 ^a^	−36.8 ± 1.5 ^c^	80 ± 4 ^bc^	3.78 ± 0.12 ^c^
EUF3C	5.68	0	NA	157 ± 4 ^f^	−16.2 ± 0.8 ^b^	NA	NA
FUF3	5.68	1.14	4.98	185 ± 6 ^d^	−37.0 ± 1.1 ^c^	81 ± 3 ^b^	13.4 ± 0.5 ^b^
UF4C	5.04	0	NA	162 ± 9 ^e^	−16.7 ± 0.9 ^b^	NA	NA
UF4	5.04	2.02	2.5	204 ± 6 ^c^	−38.1 ± 1.7 ^d^	75.8 ± 2.2 ^c^	21.7 ± 1.1 ^a^

Note: the results are means of three replicates of measurements. NA: the measurement was not applicable. The means within the same column containing different superscripts are significantly different (*p* < 0.05).

## Data Availability

All data of this study are presented within the study.

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
