# Peer review of "The Effect of pH and Sodium Caseinate on the Aqueous Solubility, Stability, and Crystallinity of Rutin towards Concentrated Colloidally Stable Particles for the Incorporation into Functional Foods"

_molecules, 2022, doi:10.3390/molecules27020534_

Round 1
Reviewer 1 Report
Dear Authors,
very interesting work. But maybe your schould keep the same format in your Figure. Specially Figure 4 (Page 10)
Best Regards
Author Response
Thanks very much for the positive feedback.
We originally submitted the figures in a separate file, and apparently, the format of the figures has changed after the publisher has inserted them into the manuscript. The format of all figures has now been unified as all figures have been replaced. In the case of Figure 4, since the pH values in this experiment are descending, the vertical axis is better to remain on the left of the graph.
Thanks again!

Reviewer 2 Report
Rashidinejad et al has carried out the research about the effect of pH and sodium caseinate on the aqueous solubility, stability, and crystallinity of rutin: towards concentrated colloidally-stable particles for the incorporation into functional foods.
Please find my comments below.
It’s not very clear from the introduction part about the importance of rutin. Is there any available clinical data to support the actual health benefits of rutin intake in human body? Authors need to address this issue in the introduction section.
Please include the structure of rutin in the manuscript.
Please use a schematic diagram to explain the mechanism behind the digestion of the rutin-NaCas complexes and rutin release in acidic PH.
From this manuscript it is not clear the interactions between NaCas and rutin. The assumption was made based on earlier docking studies; Ref 50. Authors need to discussion the interactions between NaCas and rutin with more clarity (docking studies, any other analytical technique).
What could the alternative formulation of rutin for the incorporation into functional foods.
This manuscript is acceptable for publication after major revision.
Author Response
Thanks very much for your valuable comments and insights, which have not been applied in the manuscript with the responses given under each comment as well.
It’s not very clear from the introduction part about the importance of rutin. Is there any available clinical data to support the actual health benefits of rutin intake in human body? Authors need to address this issue in the introduction section.
This was originally addressed at the beginning of the introduction, but we agree that it could be expanded. We have now addressed these health benefits in the introduction in further detail (L 41-52*). An example of a clinical trial has also been presented (L 51-3).
Please include the structure of rutin in the manuscript.
This had already been presented in the original version of the manuscript (Figure S1). Apparently, the supplementary materials have not been accessible to the reviewers in the system.
Please use a schematic diagram to explain the mechanism behind the digestion of the rutin-NaCas complexes and rutin release in acidic PH.
We think this needs a bit of clarification as the delivery system developed in this study releases rutin in the lower part of the digestion system (i.e., small intestine) with neutral pH, not the upper part with the acidic pH. We have not clarified this in the manuscript (L 621-4).
From this manuscript it is not clear the interactions between NaCas and rutin. The assumption was made based on earlier docking studies; Ref 50. Authors need to discussion the interactions between NaCas and rutin with more clarity (docking studies, any other analytical technique).
As explained in Section 3.3, the interactions between NaCas and rutin have not been reported, but we have addressed the interactions between rutin and other milk proteins and milk proteins and other flavonoids throughout the discussion (e.g., L 462-5, 501-4, 557-8, 564-8, and 574-7). As explained in the article (L 633-41), we are currently investigating the potential changes in the protein composition of samples and the structure of the protein as a function of the digestion time. Further research is also currently being carried out in our laboratory, to explore the release behavior of rutin from the rutin-NaCas particles beyond the gastric phase, as well as the possibility of using proteins (from both animal and plant sources) for the delivery of various hydrophobic flavonoids for their incorporation into various functional food products.
What could the alternative formulation of rutin for the incorporation into functional foods.
The alternative formulation for the delivery of rutin in functional food can be the co-precipitates that we have previously invented (Ref. 17) or a dried version of the delivery system presented in this paper. This has also been successfully attempted in our laboratories.
This manuscript is acceptable for publication after major revision.
Thank you again!

Reviewer 3 Report
Dear Authors,
Rutin is a challenging molecule considering its physicochemical properties and your manuscript entitled "The effect of pH and sodium caseinate on the aqueous solubility, stability, and crystallinity of rutin: towards concentrated colloidally-stable particles for the incorporation into functional foods" described a potential alternative to improve rutin characteristics to incorporate it in functional foods. Figure 3 lacked information. In lines 374-376, the pH informed was 8, although, earlier, the pH 9 was indicated. Lines 478-479 could be revised. In Conclusions, lines 624-627 could be considered too speculative. There are references with rutin and gelatin nanoparticles and microparticles that could be consulted.
Author Response
Thanks for the valuable feedback. We have addressed the comments below:
Figure 3 lacked information.
This figure was replaced. Originally, we submitted the figures in a separate file, and it appears that the formatting has changed after the publisher inserted them in the text. We apologies for this issue!
In lines 374-376, the pH informed was 8, although, earlier, the pH 9 was indicated.
At pH 9, both morphologies appeared to be present, but small crystals may start to grow at pH p, which can be in proper crystalline form at pH 8. In general, there was a dramatic decrease in rutin solubility at pH < 8.0 for both the control rutin and the rutin-NaCas mixture.
Lines 478-479 could be revised.
This has now been revised for clarity.
In Conclusions, lines 624-627 could be considered too speculative.
This was deleted.
There are references with rutin and gelatin nanoparticles and microparticles that could be consulted.
Cross-linking of protein-based products such as gelatin gels and gelatin-based delivery systems has been pointed at on Lines 587-93.
Thank you again!

Reviewer 4 Report
Additional comments:
line 42: please used the accepted name of the Sophora! Correct name: Styphnolobium japonicum!
line 58: where is the Figure S1?
line 226: Should delete „Dammak”!
line 247: how measured the „weight of particles”?
Figure 3: missed the pH values!
Table 2: please explain the abbreviation of formulations! And set the information into the Material and Methods session!
Figure 5 and 6: why the first the “B” and second the “A”?
line 599: please delete “-“ mark!
lines 624-627: should delete this sentence, this is rather speculation!
Author Response
We appreciate your time and valuable comments. Please find the respond to each comment below. The corresponding changes have also been applied in the manuscript.
Additional comments:
line 42: please used the accepted name of the Sophora! Correct name: Styphnolobium japonicum!
Corrected.
line 58: where is the Figure S1?
This is in the supplementary materials, which were submitted but apparently have not been made accessible to the reviewers. We apologies for that!
line 226: Should delete „Dammak”!
Deleted.
line 247: how measured the „weight of particles”?
Added.
Figure 3: missed the pH values!
This figure was replaced. Originally, we submitted the figures in a separate file, and it appears that the formatting has changed after the publisher inserted them in the text. We apologies for this issue!
Table 2: please explain the abbreviation of formulations! And set the information into the Material and Methods session!
Abbreviations were added and the equations were originally given in the manuscript (Eq. 1-2).
Figure 5 and 6: why the first the “B” and second the “A”?
Both figures were replaced. Originally, we submitted the figures in a separate file, and it appears that the formatting has changed after the publisher inserted them in the text. We apologies for this issue!
line 599: please delete “-“ mark!
Replaced.
lines 624-627: should delete this sentence, this is rather speculation!
Deleted.
Thanks again!

Round 2
Reviewer 2 Report
The revised manuscript has been improved as per suggestions given by reviewers. This manuscript can be accepted for publication after minor English correction.